# The North Atlantic Oscillations: Lead–Lag Relations for the NAO, the AMO, and the AMOC—A High-Resolution Lead–Lag Analysis

**Knut Lehre Seip** [1,*] **and Hui Wang** [2]

1   Department of Technology, Art and Design, OsloMet—Oslo Metropolitan University, N-0130 Oslo, Norway
2   NOAA/NWS/NCEP/Climate Prediction Center, 5830 University Research Court, NCWCP, College Park, MD 20740, USA; hui.wang@noaa.gov
*   Correspondence: knut.lehre.seip@oslomet.no

**Abstract:** Several studies examine cycle periods and the interactions between the three major climate modes over the North Atlantic, namely the Atlantic meridional overturning circulation (AMOC), the Atlantic multidecadal oscillation (AMO), and the North Atlantic oscillation (NAO). Here, we use a relatively novel high-resolution Lead–lag (LL) method to identify short time windows with persistent LL relations in the three series during the period from 1947 to 2020. We find that there are roughly 20-year time windows where LL relations change direction at both interannual, high-frequency and multidecadal, low-frequency timescales. However, with varying LL strength, the AMO leads AMOC for the full period at the interannual timescale. During the period from 1980 to 2000, we had the sequence NAO→AMO→AMOC→NAO at the interannual timescale. For the full period in the decadal time scale, we obtain NAO→AMO→AMOC. The Ekman variability closely follows the NAO variability. Both single time series and the LL relation between pairs of series show pseudo-oscillating patterns with cycle periods of about 20 years. We list possible mechanisms that contribute to the cyclic behavior, but no conclusive evidence has yet been found.

**Keywords:** climate; ocean oscillations; AMOC; AMO; NAO; Lead–lag relations

## 1. Introduction

There are three climatic indices that characterize the variations of the North Atlantic Ocean: the North Atlantic oscillation (NAO), the Atlantic multidecadal oscillation (AMO), and the Atlantic meridional overturning circulation (AMOC). All three indices show cycle periods, e.g., Seip and Gron [1], but there is no consensus on the mechanisms that cause specific cycles [2] (p. 240). Neither are there any established mechanisms that govern lead and lag relations that are found between the three, although interactions between the AMO and the NAO have been suggested as the cause for quasi-periodic multidecadal variability in the NAO [3] (p. 2084). Here, we examine the interactions between them in terms of high-resolution Lead–lag (LL) relations.

Traditional LL analysis, for example, the cross-correlation analysis, requires long time series (describing, e.g., more than three peaks in the cyclic series), whereas the LL method we use calculates LL relation over three consecutive observations in the paired time series. Confidence intervals can be calculated for nine consecutive observations, but under certain conditions for the distribution of observations.

We would have liked to add and discuss the mechanisms that would generate the LL patterns we identify. For LL patterns that are persistent across the study period, possible mechanisms have been discussed in the literature, but for LL patterns that change with time, few—if any—suggested mechanisms exist.

Since the LL method is relatively novel and distinguishes itself from (most) traditional LL analyses both in calculation and in interpretation, we here shortly sketch characteristics

of the method using paired sine-function series with a common cycle period (λ) and with a phase shift (δ) between them as examples. The high-resolution LL method defines a leading series as a series where the peak of the leading series is less than λ /2 before the lagging series. A lagging series is less than λ/2 after the leading series. Thus, the two series are either leading or lagging each other. A leading series that is inverted will become a lagging series, emphasizing that the units for measuring the series are important. For example, if both series are measured as temperatures, both series should define high temperatures as positive. If the series are measured in different units, the context in which LL relations will be interpreted is important. In this example we refer to LL relations between peaks, but the LL relations apply to all parts of the paired series longer than three consecutive observations.

A particular feature comparing the results using the high-resolution LL method with results from, e.g., cross-correlation methods is that if LL relations change over short time windows, e.g., ten observations, then cross correlation may show no significant LL relations (the short-term LL relations cancel out), but significant LL relations are still present.

## 2. Materials

We use the time series of the three indices: the AMOC, AMO, and NAO.

The NAO measures the sea-level pressure difference between a Southern station, Lisbon, and a Northern station, Reykjavik. We use the annual NAO index from 1947 to 2020 obtained from https://psl.noaa.gov/gcos_wgsp/Timeseries/ (accessed on 15 February 2022).

In the following, we will use the term 'ocean oscillation' for the NAO even though it measures sea-level pressure differences. However, the sea-level pressure differences translate into wind directions and to temperature differences in the ocean, in particular during annual fall–winter cooling periods since 2003 in the North Atlantic [4] (p. 8111). The positive (+) NAO drives warmer, wetter conditions in northern Europe [5].

The AMO is measured as fluctuations in the North Atlantic sea-surface temperature (SST) anomalies, 0–70° N. We obtained the AMO series from http://www.esrl.noaa.gov/psd/data/timeseries/AMO/ (accessed on 15 February 2022).

The AMOC is a system of currents in the Atlantic Ocean. It shows a northward flow of warm saltwater in the upper layers of the Atlantic, including the Gulf Stream, and a southward flow of colder, deep water, which is part of the thermohaline circulation. Its characteristics are examined further by Caesar [6–8]. The Caesar series has been compared to proxies from about 40° N to 60° N, such as coral food sources in $\delta^{15}$ N [6]. The data were obtained from Levke Caesar, Potsdam Institute for Climate Impact Research. The AMOC is the variable that most directly expresses transfer of heat (down to 2000 m, National Oceanography Centre, Available online: https://www.rapid.ac.uk/rapidmoc/ (accessed on 15 February 2022). The AMOC has been instrumentally observed at 26° N since 2004, the Rapid Climate Change Programme, RAPID. We compare two versions of the AMOC time series in Appendix A and we examine our choice of AMOC series in the discussion section. The AMOC series have unit Sv.

The Ekman transport is considered as a portion of AMOC, and the data are from Wang et al. (2019), covering the period from 1990 to 2015. It should be noted that the Ekman transport is mostly driven by wind and the unit is Sv.

## 3. Methods

We use a relatively novel technique for calculating Lead–lag (LL) relationships between cyclic time series [9]. The method relates to a dual representation of the time series, x(t) and y(t), first as series depicted as a function of time and second as depicted in a phase plot with one series on the x-axis and the other series on the y-axis. Time in the phase plot is then shown as the trajectories between points. For a quick intuitive illustration, see: https://en.wikipedia.org/wiki/Lissajous_curve#/media/File:Lissajous_phase.svg (accessed on 15 February 2022).

The method is described in detail in Seip and Grøn [10], but recently Krüger [11] has described a LL method that is based on wavelet techniques and on the same dual representation of paired time series.

A sine function (red curve) that is to the left of the target sine function (black curve) is a leading series (Figure 1a). With the target series on the x-axis and the leading series on the y-axis the trajectories in the phase plot rotate clockwise (red curve, negatively per definition) (Figure 1b). Correspondingly, with the target series on the x-axis and the lagging series on the y-axis, the trajectories rotate counterclockwise (blue curve, positive per definition). Thus, we can identify LL relations by the way that trajectories rotate in the phase space. For time series normalized to unit standard deviation, the trajectories will form an ellipse-like curve with the major axis either in the 1:1 direction or the −1:1 direction. For shifted perfect sine functions with common cycle periods, the minor axis will show the phase shifts between the sine functions.

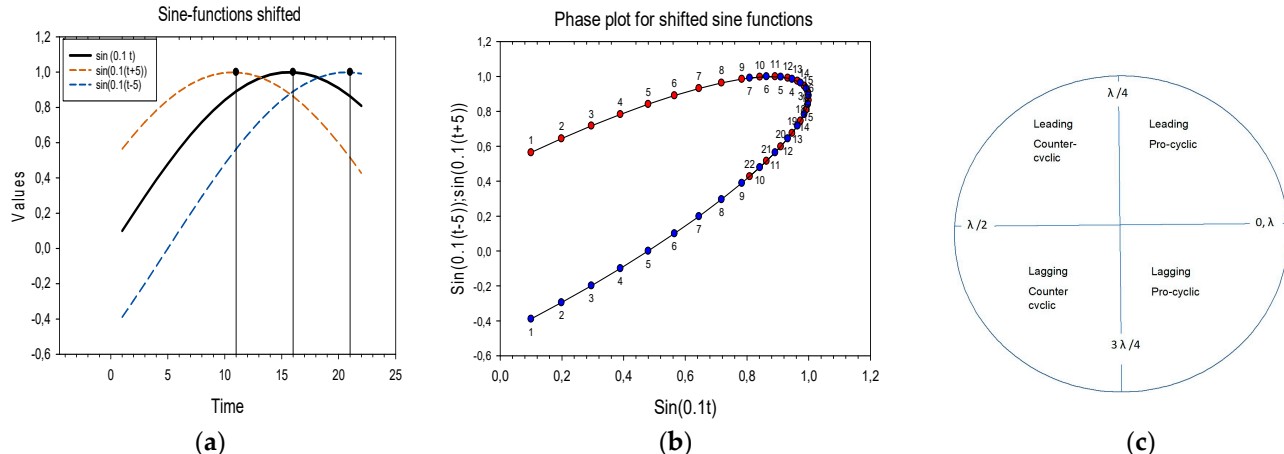

**Figure 1.** Method explanation. (**a**) Part of sine functions y = sin (0.1 (t + δ)) where δ is −5 (red curve) and +5 (blue curve). The black, bold curve is the target cure that we want to examine LL relations for t = 0 to 25. (**b**) Phase plot for pair of curves. Red curve shows the leading sine function, and the blue curve shows the lagging sine function. (**c**) The LL relation and the correlations between cyclic time series with common cycle period, λ, as a function between the phase shifts, δ, between them.

The rotational direction represented by the angle θ between two successive vectors $v_1$ and $v_2$ through three consecutive observations is calculated from Equation (1):

$$\theta = \text{sign}(v_1 \times v_2) \cdot \text{"Arccos"} ((v_1 \cdot v_2) / |v_1| |v_2|) \tag{1}$$

The vectors are calculated as $(y_i - y_{i-1})/(x_i - x_{i-1})$ with i = 2, 3, . . .
We define a measure of LL strength as

$$LL = (N+ - N-)/(N+ + N-) \tag{2}$$

where N+ and N− is the number of positive and negative angles, θ, in a set of *n* consecutive observations in the two series. Using *n* = 9 and with N+ = 9 and N− = 0, we obtain LL = (9 − 0)/(9 + 0) = 1. The number 9 is a trade-off between the goal of identifying LL relations for short time windows and the goal of identifying a confidence band. In the time series mode, it means that one series leads the other for nine consecutive observations. In the phase representation, it means that when the two series are plotted as trajectories in the phase plot, the trajectories will rotate persistently in one direction.

*Confidence interval.* The 95% confidence interval (CI) is based on the probability that two uniformly stochastic series will show a persistent rotation in one direction. It is calculated with Monte Carlo simulations applying Equation (1) and Equation (2) to two uniformly stochastic series, and the confidence limits are the asymptotic values for 1000 replicates. Values of LL < −0.32 and LL > 0.32 suggest that for time series longer than nine time

steps, the LL values are significant at the 95% level. When time series are smoothed with a smoothing algorithm, the probability that consecutive angles will have the same sign increases, so the CI does not strictly apply to smoothed series. However, by comparing LL relations for smoothed series with LL relations for unsmoothed series, the confidence in the LL relations for the smoothed series may be enhanced.

*Cycle periods*. We determine cycle periods by two methods, by the power spectral density (PSD) algorithm that is available in most statistical packages, and by calculating cumulative angles for the trajectories in the paired cyclic-series phase plot (Figure 1b). If the trajectories close, the number of time steps required for closing corresponds to one cycle period in the time-series plot. However, two stochastic series would have a probability of $p > 0.05$ for showing cycle periods shorter than about seven time steps. Thus, cycle periods fewer than seven time steps long could be the result of stochastic variability. This method for calculating cycle periods will be called the cumulative angle method.

*Phase shifts, or Lead–lag times*. If two sine series with a common cycle period, λ, coincide perfectly, the ordinary linear regression (OLR) for the cycles would show a regression coefficient β = 1.0 and r = 1.0. If one series is displaced $\frac{1}{4}$ λ relative to the other, the two sine functions would show a perfect circle and β = 0 and r = 0. An approximation to phase shift (PS) or the lead or lag time can be calculated as

$$PS \approx \lambda/2\pi \times (\pi/2 - \text{Arcsine}(r)) \tag{3}$$

To calculate the PS, we must know the cycle period in advance. We calculate two versions of the PS, one based on the running average λ, over five time steps, and one based on the average λ for the cycle periods identified for the full series.

Pro-cyclic and counter-cyclic relations. If the OLR β-coefficient is positive, we say that the two series are pro-cyclic. If the β-coefficient is negative, we say that it is counter-cyclic. Figure 1c shows that a LL relation can be positive both for pro-cyclic and counter-cyclic series. With counter-cyclic series, the PS is normally larger than with pro-cyclic time series.

*Smoothing*. We use the LOESS-smoothing algorithm. The algorithm has two parameters: the parameter (f), which shows how large fraction of the series that is used as a moving window; and the parameter (p), which shows the polynomial degree used for interpolation. We always use $p = 2$. With 75 years of observations and f = 0.2, the moving time window is 15 time steps and the smoothing results for seven years at the beginning and the end of the time series are supported by less than 15 years of information. We use the LOESS-smoothing algorithm as implemented in SigmaPlot, but the algorithm is implemented in many statistical packages and gives the same results. Since we always use the parameter $p = 2$, we use the nomenclature LOESS(f) for LOESS smoothing.

An example showing the LL relations, common cycles, and phase shifts for the AMOC LOESS(0.3) and the same series shifted six time steps forward are shown in Appendix B. Both the data and the calculations of LL relations are available from the author. The LL calculations are made with essentially only one column in an Excel spreadsheet, Equation (1).

## 4. Results

Variations of the North Atlantic climate have been shown at interannual and multi-decadal time scales, e.g., Wang et al. (2015) [12], Yashayaev et al. (2016) [4], among many others. An emerging picture is that the observed variations are not due to single processes but several processes, and each process has a distinct mechanism and time scale [13]. Figure 2a shows the AMOC, AMO and NAO in raw format and LOESS(0.3)-smoothed. Figure 2b shows power spectral density graphs for the three series. The raw PSD graphs have been normalized to unit standard deviation. The droplines show the most pronounced peaks in the graphs.

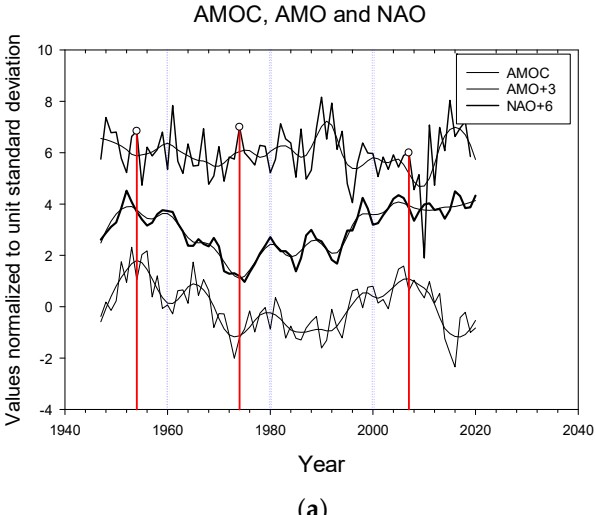
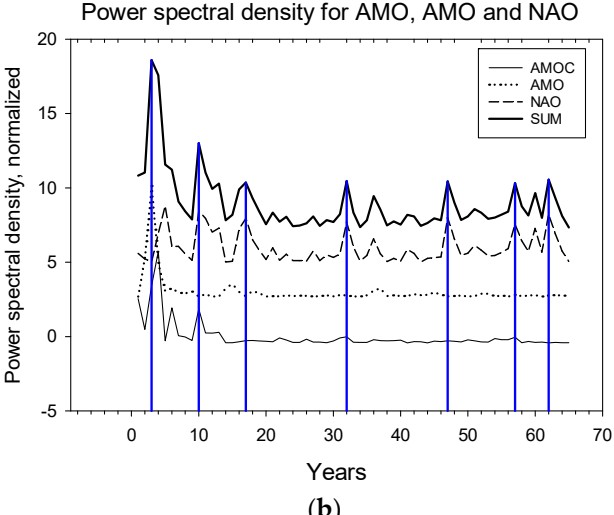

**(a)**  **(b)**

**Figure 2.** AMOC, AMO and NAO. (**a**) Time series raw and LOESS(0.3)-smoothed. The detrended and LOESS(0.3)-smoothed versions of AMOC shifts sign from starting with (+) in 1947, then 1969, 1997, 2010. AMO starting from (+) in 1947, 1996, 1999. NAO starting from (+) in 1947, 1952, 1972, 1997, 2014, 2020. (**b**) Power spectral density. Peaks at 3, 10, 17, 32, 47, 57, and 62 years.

*LL Relations, Cycle Periods, and Phase Shifts*

We first present graphs that show results for the three pairs that can be constructed between AMOC, AMO, and NAO. The presentation follows same configurations for all three pairs. In addition to time-series plots, we show a phase plot for a section of the time series from the year 2000 to 2010. The phase plot gives two types of information: it shows an ordinary linear regression (OLR) between the two series (the β-coefficient), and it indicates which of the two series is leading the other (the rotational direction).

*AMOC versus AMO.* The two time series are shown in Figure 3a. The upper zigzag curve indicates the length of cycle periods defined by the cumulative angle method. Cycle periods range from 1 to 13 years with an average of 5.6 years. The LL relation between AMOC and AMO shows that AMO basically leads AMOC, and significantly during the periods 1980–1990 and 2000–2020 (Figure 3b). Phase shifts and the β- coefficients between the two time series are shown in Figure 3c. The β-coefficient is positive 63% of the time.

A phase plot for a section of the LOESS(0.1)-smoothed time series from the year 2000 to 2010 shows that in the first part of the series, from 2000 to 2005, the trajectory rotates clockwise (−negative) and AMOC lags AMO, then there is a period from 2005 to 2008 where rotation is counterclockwise (+positive) and AMOC leads AMO. For the last two years, rotation is clockwise (-) and AMOC lags AMO (Figure 3d). The rotational directions in Figure 3d correspond to the positive/negative black bars in Figure 3b.

We LOESS(0.3)-smoothed the series to disentangle decadal and multidecadal variability in the series. For the time series 74 time steps long, an f value of 0.3 corresponds to 22 time steps. The smoothed series are shown in Figure 3e (and can be compared with the raw series in Figure 4a). Cycle periods now range between 15 and 26 years with an average of 17.25 years. AMOC LOESS(0.3) now leads AMO LOESS(0.3) most of the time, except during the period from 2000 to 2018. The average lead time is 1.4 to 2.1 years (Two calculation methods, standard deviation for both ≈ 2 years).

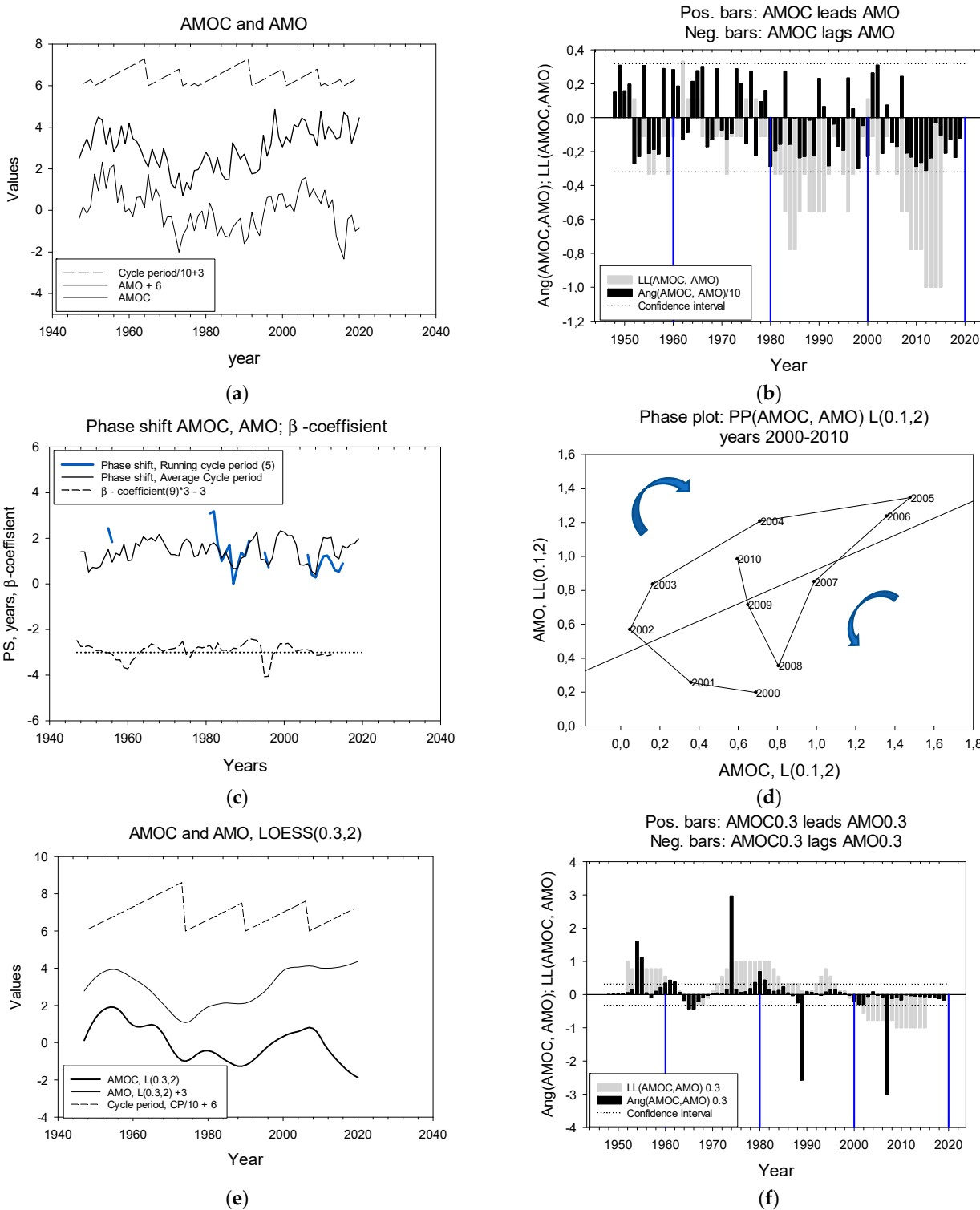

**Figure 3.** LL relations for AMOC and AMO. (**a**) Time series raw. The upper zigzag curve shows cycle periods. (**b**) LL relations between AMOC and AMO. Black bars show angles, θ(3), calculated over three consecutive observations; grey bars show the sign of angles, LL(9) calculated over 9 years. Dashed lines show confidence interval. Droplines designate time windows of 20 years starting at 1960. (**c**) Phase shift calculated with two methods and the β-coefficient for the two series. (**d**) Phase plot for the two series with AMOC and AMO slightly LOESS(0.1)-smoothed on the x- and y-axis, respectively. (**e**) The two series LOESS(0.3)-smoothed and their cycle periods. (**f**) LL relations between the two series. Black bars for θ(3), and grey bars for LL(9).

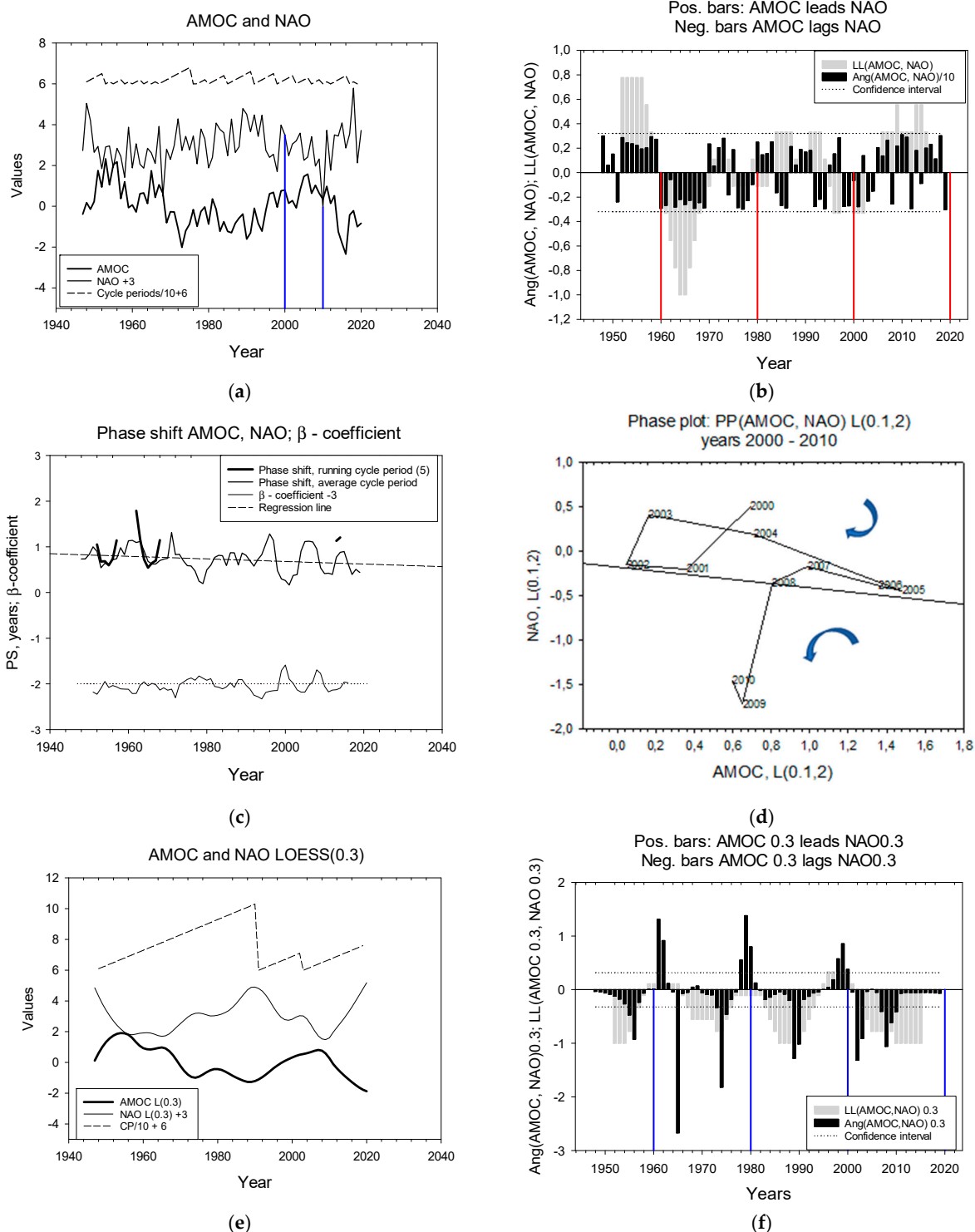

**Figure 4.** LL relations for AMOC and NAO. (**a**) Time series raw. The upper zigzag curve shows cycle periods. (**b**) LL relations between AMOC and NAO. Black bars show angles, θ(3), calculated over three consecutive observations; grey bars show the sign of angles, LL(9) calculated over 9 years. Dashed lines show confidence interval. Droplines designate time windows of 20 years starting at 1960. (**c**) Phase shift calculated with two methods (Thin line: average cycle period; bold line: running average cycle period) and the β-coefficient for the two series. (**d**) Phase plot for the two series with AMOC and NAO slightly LOESS(0.1)-smoothed on the x- and y-axis, respectively. (**e**) The two series LOESS(0.3)-smoothed and their cycle periods. (**f**) LL relations between the two series. Black bars for θ(3), and grey bars for LL(9).

*AMOC* and *NAO*. The two time series are shown in Figure 4a. Cycle periods range from 1 to 8 years with an average of 2.9 years. The series lead and lag each other for almost equally long periods (Figure 4b). Phase shifts and the β-coefficients between the two time series are shown in Figure 4c. The phase plot (Figure 4d) for a section of the time series from the year 2000 to 2010 shows that in the first part of the series, from 2000 to 2005, the trajectory rotates clockwise (-) and AMOC lags AMO, then there is a period from 2006 to 2009 when the rotation is counter clockwise (+) and AMOC leads NAO. We LOESS(0.3)-smoothed the series to disentangle decadal and multidecadal variability in the series. The smoothed series are shown in Figure 4e. Cycle periods range between 11 and 43 years with an average of 23 years. AMOC LOESS(0.3) lags NAO LOESS(0.3) most of the time, Figure 4f. The average lead time is 7.6 to 9.5 years. (Two calculation methods, standard deviation for both ≈ 2.6 years).

*AMOC* and *NAO*. The two time series are shown in Figure 5a. Cycle periods range from 1 to 7 years with an average of 2.3 years. The series lead and lag each other for almost equally long periods (Figure 5b and Table 1). Phase shifts and the β-coefficients between the two time series are shown in Figure 5c. A phase plot for a section of the time series from the year 2000 to 2010 shows that in the first part of the series, from 2000 to 2003, the trajectory rotates counterclockwise (+) and AMO leads NAO, then there is a period from 2003 to 2007 when the rotation is counterclockwise (+ positive) and AMO lags NAO. For the last period, from 2007 to 2010, AMO again leads NAO. The LOESS (0.3)-smoothed series are shown in Figure 5e. Cycle periods now range between 10 and 44 years with an average of 23 years.

**Table 1.** Characteristics of Lead–lag relations between pairs of North Atlantic climate modes. The bold numbers in the two last columns suggest that for the corresponding pairs, identifying LL relations with cross-correlation techniques may give results that are similar to ours.

| Series 1 | Series 2 | Smoothing Parameter, *p* | Cycle Period Years | | Phase Shift | LL(1,2) | β-Coefficient |
|---|---|---|---|---|---|---|---|
| | | | average | max | years | % counter clockwise | % positive |
| AMOC | AMO | 0 | 5.6 | 13 | 1.4–1.5 | **36** | 68 |
| | | 0.3 | 17 | 26 | 1.6–2.1 | 56 | **82** |
| AMOC | NAO | 0 | 2.9 | 8 | 0.78–1.07 | 56 | **34** |
| | | 0.3 | 23 | 43 | 7.6–9.5 | **25** | 7 |
| AMO | NAO | 0 | 2.3 | 7 | 0.77–1.3 | 50 | **5** |
| | | 0.3 | 2.3 | 44 | 6.5–7.1 | **33** | 32 |
| Ekman | NAO | 0.2/0.1 | 4 | 5 | 0.7–0.8 | 60 | **80** |

AMO LOESS(0.3) lags NAO LOESS(0.3) most of the time (Figure 5f) with a lag time of 6.5 to 7.1 years (lag times not shown). The average lead time is 6.5–7 years. (Two calculation methods, standard deviation for ≈ 4 years).

The results shown in Figures 3–5 are summarized in Table 1. We use principal component analysis (PCA) to show the relations between the six raw and the LOESS(0.3)-smoothed LL relations in Figure 6a. For the AMOC/NAO and the AMO/NAO high- and low-frequency pairs, the LL relations are inversely related. The relations between the six β-coefficients show that they are inversely related for the AMOC/AMO and the AMO/NAO pairs, Figure 6b. For all six LL relations there appears to be common time windows of about 20 years where LL relations are persistent in one direction, ≈1960–1980, ≈1980–2000, and ≈2000–2020 (Figure 6c).

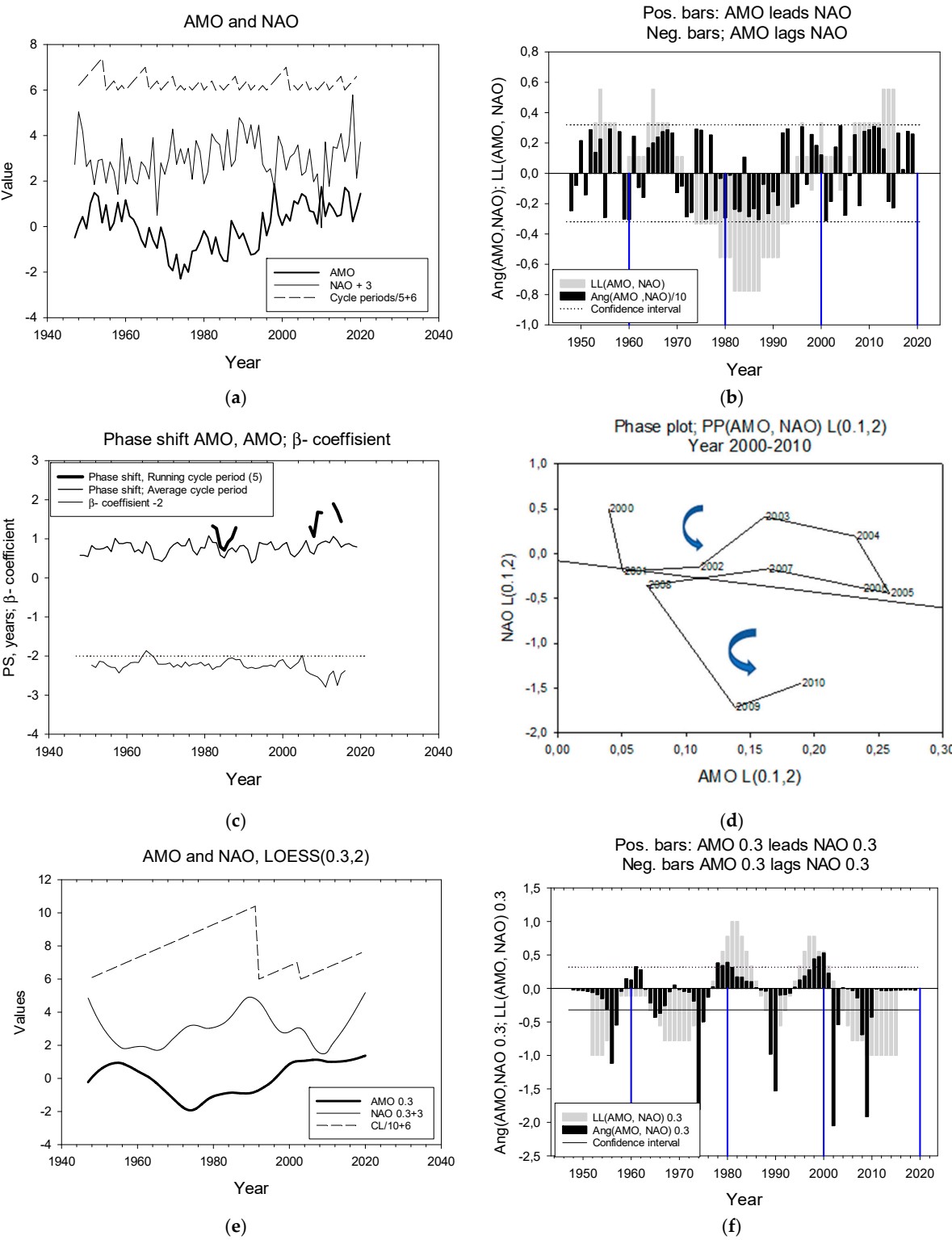

**Figure 5.** LL relations for AMO and NAO. (**a**) Time series raw. The upper zigzag curve shows cycle periods. (**b**) LL relations between AMO and NAO. Black bars show angles, θ(3), calculated over three consecutive observations; grey bars show the sign of angles, LL(9) calculated over 9 years. Dashed lines show confidence interval. Droplines designate time windows of 20 years starting at 1960. (**c**) Phase shift calculated with two methods (Thin line: average cycle period; bold line: running average cycle period) and the β-coefficient for the two series. (**d**) Phase plot for the two series with AMO and NAO slightly LOESS(0.1)-smoothed on the x- and y-axis, respectively. (**e**) The two series LOESS(0.3)-smoothed and their cycle periods. (**f**) LL relations between the two series. Black bars for θ(3), and grey bars for LL(9).

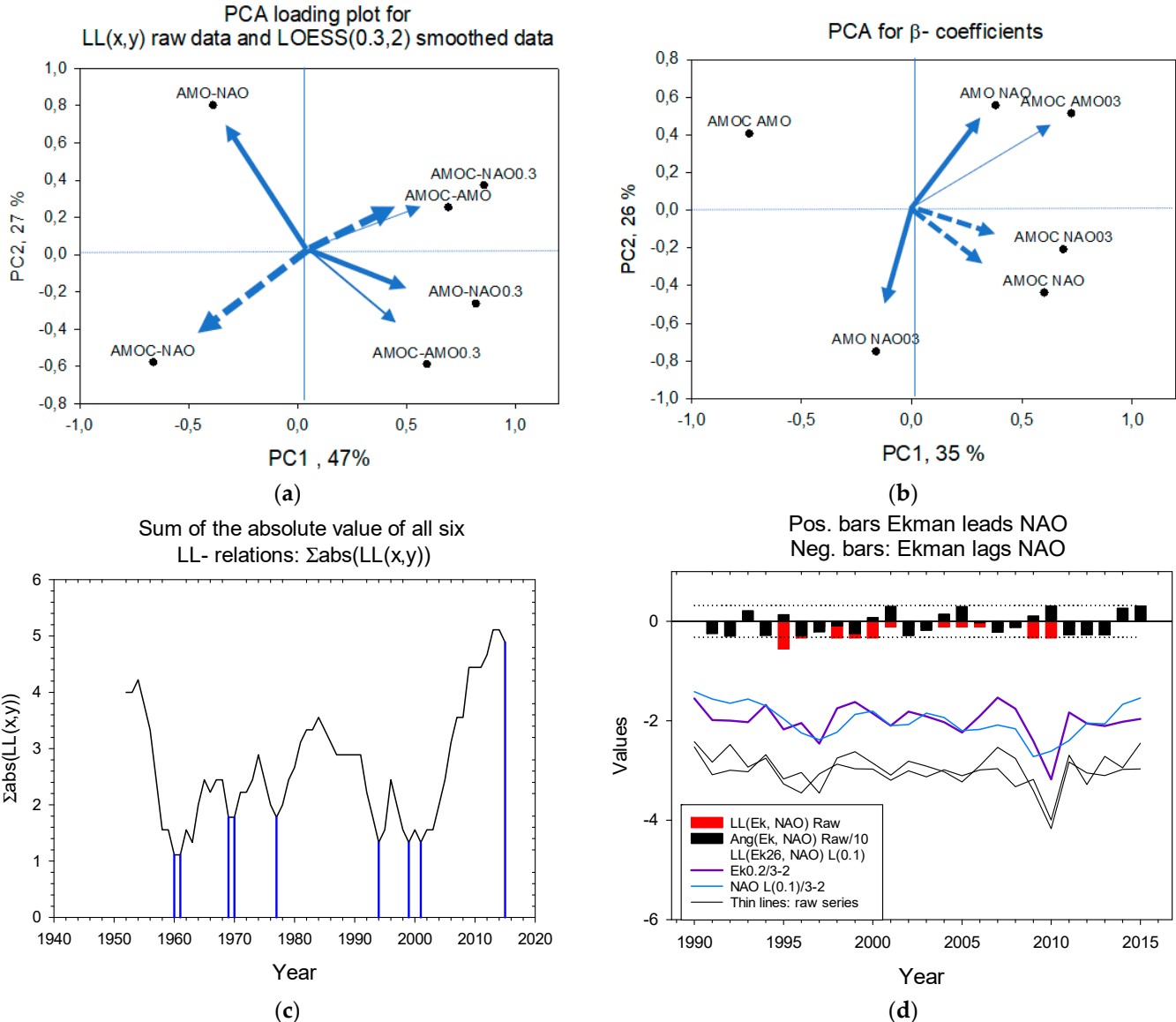

**Figure 6.** Summary characteristics. (**a**) Principal component loading plot for LL relations between the three modes, AMOC, AMO, and NAO, their high- and low-frequency series. (**b**) PCA loading plot for the series β-coefficients (slopes between paired series), their high- and low-frequency components. (**c**) The stacked sum of the absolute value of the LL (x,y) values. The droplines designate troughs in 1960–1961, 1968–1969, 1977, 1994, and 1999–2001. (**d**) Ekman time series, raw series (dashed lines) and Ekman 26ʹ LOESS (0.2)-smoothed, and NAO LOESS(0.1)-smoothed (Full lines). The red bars show LL relations based on the raw data, and the light blue bars show LL relations based on LOESS-smoothed series.

*The Ekman transport and NAO.* Figure 6d shows two versions of the Ekman and the NAO time series from 1990 to 2014. The series show Ekman LOESS(0.2)-smoothed and NAO LOESS(0.1)-smoothed. The reason we use LOESS(0.2) for the Ekman series is that the LOESS algorithm did not accept LOESS(0.1) smoothing for this short series. The LL results depicted with black bars (angles) and red bars (LL relations) show that LL (Ekman, NAO) gave nonsignificant results. However, with slight smoothing, Ekman appears to lead NAO during the period from 1998 to 2007, but smoothing enhances the probability that (pseudo)-significant results appear. Cycle periods are 4–5 years and phase shifts are less than one year; that is, less than the series resolution. Therefore, it is most likely that there is no measurable time lag between Ekman and NAO variability (Ek = 0.533 × NAO + 0.0, R = 0.53,

$p = 0.005$, $n = 26$), and this is consistent with Wang and Brickman [14], which shows that the NAO is a good indicator for the Ekman driven portion of the AMOC.

In Figure 7, we summarize the LL relations for the three 20-year periods for the high-frequency time series (upper three graphs) and for the low-frequency series (lower three graphs), respectively. Thickness of arrows suggests strength of LL relations.

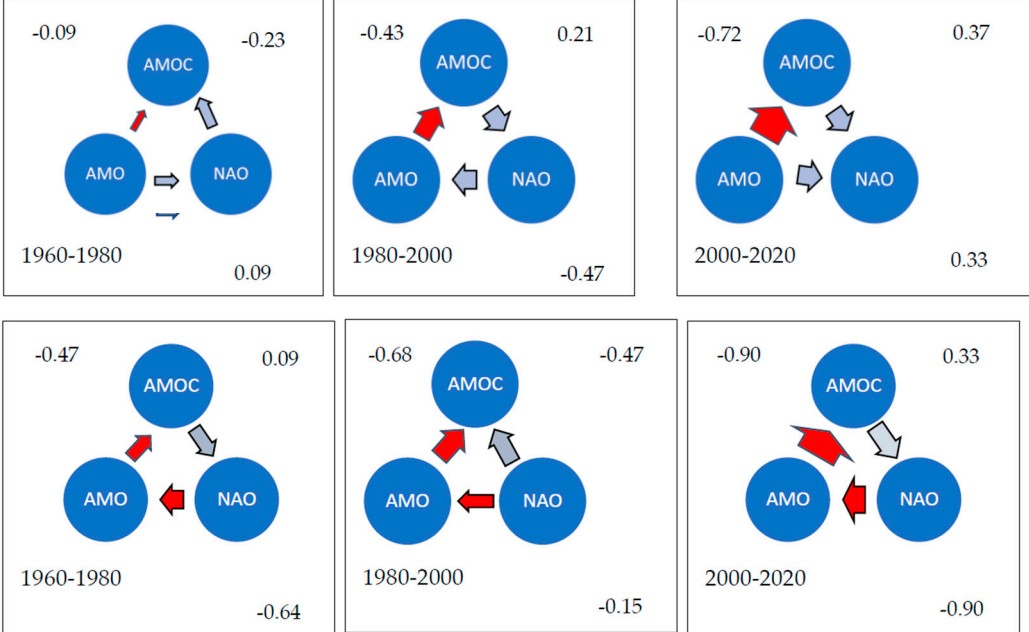

**Figure 7.** LL relations between the AMOC, the AMO, and the NAO: LL(AMOC, AMO), LL(AMOC, NAO), LL(AMO, NAO). AMOC is measured in Sverdrup, Sv; AMO is measured in temperature degrees; and NAO is measured in sea-level pressure, hPa. We have distinguished three periods corresponding to the LL patterns in Figure 2. The arrows indicate leading directions, and arrow thickness indicates leading strength, Equation (2). The numbers in each corner show numerical values for direction (LL(x, y) > 0: x→y. The leading strength is the average for the ten mid-years in each period. *Upper row*: high-frequency oscillations. *Lower row*: low-frequency oscillations. Red arrows suggest that the leading role is persistent across all time periods.

For high-frequency oscillations, cycle periods are 2–6 years and Lead–lag times are about 1 year.

For low-frequency oscillations, cycle times are 17–23 years on the average, but there are long cycles of 43–44 years. Lead–lag times are 6.5–9.5 years for the AMOC/NAO pair and the AMO/NAO pair, but 1.6–2.1 years for the AMOC/AMO pair.

## 5. Discussion

We first discuss cycle periods in the ocean time series and in the LL series obtained from the paired series. Then, we discuss the LL relations and the β-coefficients that we found for each of the three pairs. The LL relations were calculated separately for the high-frequency component based on annual data and the low-frequency component based on LOESS(0.3)-smoothed data. The LOESS(0.3) smoothing corresponds to a running window length of 22 years. In the literature, a series of filtering and smoothing techniques have been used to identify the low-frequency component of raw ocean variability series. Wills and Schneider [15] (p. 2488) applied a Lanczos filter with 10 years low-pass cutoff to identify the low-frequency component of the Pacific decadal oscillation (PDO) and the El Niño Southern Oscillation (ENSO), and Nigam and Sengupta [16] used a LOESS(0.1) filter, but did not distinguish between high- and low-frequency versions of the time series. Wang and Zhang [14] used a cutoff of 7 years to define high and low frequencies. They found that high-frequency changes of the sea-surface height (SSH) were mostly driven by wind

and that the SSH contributes to barotropic currents of the Atlantic Ocean. We will discuss the choice of ocean variability series in the robustness section.

*5.1. Cycle Periods*

The high-resolution LL method we use allows us to see changes in LL relations over both decadal and shorter time intervals. For both the raw annual AMOC, AMO, and NAO data and the LOESS (0.3)-smoothed decadal data, there are time windows of approximately 20 years among which the LL relations change directions. This 20-year period can be compared to the cycle periods identified by the PSD method in the AMOC, AMO, and NAO time series and with the cycle periods we identify in the decadal, LOESS(0.3)-smoothed series.

In the power spectral density graph (Figure 2b), the NAO time series shows a 17-year cycle period, but both the NAO and the AMOC show 10-year cycle periods. Rózynski' [17] fond an 8-year oscillation in the winter NAO series. Ten-year cycles appear to also be a component of time series for PDO, ENSO, and the Ninõ 3.4 (Wills et al. [15] (p. 2490)). Thus, decadal periods seem to be prevalent in ocean oscillations. In addition to cycle periods at decadal and bidecadal scales, there are also longer cycle periods on multidecadal scales, 40 to 60 years (Rózynski [17] (p. 117), Nigam and Sengupta [16] (p. 5487), and Sun and Jin [3] (p. 2087) and references therein).

*Disentangling time-series components.* There are several methods for disentangling components of an observed time series under the assumption that the series is a superposition of different series that each represents a separate cycle-generating mechanism (Capotondi and Wittenberg [18]). Furthermore, there are different criteria for when one series is uniquely distinguished from another, e.g., Huang and Wu [19]; Wills and Schneider [15]. However, there are no canonical criteria because there will always be the chance that there exists dynamic chaos in the system (May, Figure 2.2, [20]), and two independent processes may interact to produce a spurious peak that does not "belong" to any of the original series [21]. Here, we identify low frequency series as the result of LOESS(0.3) smoothing.

To our knowledge, there is no consensus on what mechanisms cause ocean variability patterns. Factors that cause persistent lead or lag times between ocean variability series are discussed abundantly in the literature, but to our knowledge, factors that cause LL relations to change are not discussed to any extent.

*Exploring the possible* factors that contribute to climate time-series periodicity is hampered by the relatively short records and high uncertainty in in climate data before, e.g., 1880. We can envisage four categories of mechanisms that could generate stationary (pseudo-)cycle periods: *First*, cycles would arise when a "compartment" in the cycle system becomes saturated and mechanisms start to act in the opposite direction. For example, cycle periods have been suggested to be caused by feedback mechanisms among pseudo-cyclic ocean and atmospheric mechanisms, e.g., Sun et al. [3] (p. 2086) and Dai et al. [22] (p. 557) and references therein. Wang and Brickman [14] suggest that the low-frequency AMOC may relate to hydrographic changes in the Labrador Sea deep layers, but Li and Lozier [23] are skeptical to the role of the deep western boundary changes for the North Atlantic overturning characteristics. *Second*, dynamic chaos and bifurcation would cause cycles [24]. Arzel et al. [25] (p. 6418) suggest that basin mean kinetic energy density and a bifurcation give a period of about 22 years. *Third*, the interaction between stochastic variabilities may cause distinct cycles to appear [26]. *Fourth*, there are several external mechanisms that may act as control knobs for processes that cause cycles (or amplify processes that cause cycles), e.g., the solar cycles [27], volcanisms [28] and volcanic aerosols [29–31], and oceanic tides [32,33].

To our knowledge, there are few techniques that are used in climatology for assessing the contribution from the possible mechanisms that cause pseudo-cyclic variability. Methods based on leading relationships (the cause has to come before the effect) are discussed in Sugihara et al. [34], Krüger [11], and Seip and McNown [9]. A second method is based on the improvement in forecasting skill, the Granger cause method (Granger [35]). If dy-

namic chaos or bifurcation is responsible for pseudo-cyclic periods, interpretations in terms of ocean or atmospheric processes are not straightforward but would require modeling support, like in [36]. For the other mechanisms, physical reasoning should support the numerical results.

### 5.2. The AMOC–AMO Relations

The AMOC–AMO relations. The *high-frequency* component of the AMOC/AMO pair showed that AMOC generally lagged AMO for the period from 1947 to 2021, but more persistently after 1980 than before. The high-frequency variabilities may be due to wind forces. However, since there appears to be persistent LL relations, the wind forcing of the AMO may impose a cyclic behavior on the high frequency of the AMOC. The lead time for the AMO is between 1 and 2 years, suggesting that this is the time required to overcome the thermal inertia associated with slow oceanic processes at this frequency.

The *low-frequency* component of the AMOC/AMO pair shows that AMO leads AMOC. Thus, AMO leads both the high- and the low-frequency component of the AMOC. The cycle periods for the low-frequency component are now on average 17 years and the lead times about 2 years. The β-coefficient is largely positive, suggesting that AMO and AMOC moves in concert (Figure 3c).

The low-frequency component of the AMOC is probably weakened by increases in buoyancy in the Greenland–Iceland–Norwegian Seas (Armstrong and Valdes [37] (p. 2498) and Li and Sun [38]) or by deep-water formation in the subpolar North Atlantic (Wang and Li (2021)).

### 5.3. The AMOC–NAO Relations

*High-frequency oscillations.* We show that AMOC both leads and lags the NAO. For the AMOC/NAO pair we only found significant LL relations in the beginning of the study period from 1950 to 1970 (Figure 4). The time lag is about one year. The β-coefficient for the two series is positive 34% of the time. With only winter months for the NAO (D, J, F), the period where AMOC lags NAO lasts longer, from 1961 to 1988, the mid-period from 1980 to 2000 shows no distinct LL relations, but from about 1990 AMOC leads NAO (Figure A1 in Appendix A). The annual time series (Figure 4), and the seasonal series (Figure A1) do not show dramatically different LL relations, but the leading role of the NAO becomes more pronounced when the time series are restricted to winter values. Wang and Brickman [14] (p. 10) found a regime shift in the AMOC between the 1990–2000 and the 2001–2015 period that they suggest is due to a westward shift in the NAO winter (J,F,M) centers of action. Our results would indicate that the regime shift started earlier than 2000, but that it became more pronounced around 2000.

*The low-frequency oscillations* show one long cycle period of 43 years, and one short period (10 years) very similar to the pattern of the AMO/NAO pair to be discussed below. The lead time is about 6.5–9.6 years for all cycle periods. The LL relations show that NAO leads AMOC over 75% of the time. The β-coefficient is negative 95% of the time. The LL relations for the low-frequency series are the opposite of the LL relations for the high-frequency series (Figure 6a (dashed lines)).

The dominating clockwise rotation, NAO leading AMOC, for the low-frequency AMOC and NAO suggests that we can compare our results with the results of Sun and Li (p. 2, Figure 10, [3]). They address cycle periods of 50–70 years in the NAO and the AMOC (21 years running mean of both series) and find that the NAO has a positive forcing effect on the AMOC (leading with 15–20 years), but that AMOC "in turn" has negative feedback on the NAO; that is, causing a negative NAO (-) phase. The authors found that these positive and negative feedback mechanisms act successively, leading to quasi-periodic multidecadal variability in the NAO and AMOC series. However, there is no explanation for why the mechanisms cause the cycle periods that are observed (50–70 years). The authors treat AMO and AMOC together, using the nomenclature "AMOC/NAO +" for the positive

phases of the two time series. This seems reasonable since the two series correlate well, β (average) = 0.5.

### 5.4. The AMO–NAO Relations

*High-frequency oscillations.* We show that the AMO both leads and lags the NAO, shifting LL direction about every 20 years (Figure 5). The short cycle periods of 2–3 years suggest that the variabilities are wind-driven. However, since there is persistent Lead–lag relations for 20 years, the wind forcing of the one series may impose a cyclic behavior on the other series. The lead time for the AMO is around 1 year, suggesting that this is the time required to impose the high-frequency cyclic behavior between NAO and the AMOC series. The β-coefficient is strongly negative, suggesting that the two series are out of phase (Figure 5c).

*Low-frequency oscillations.* The NAO leads AMO. There is one long cycle period of about 44 years, one short period (10 years), and one unfinished period. The lead time is around 9.5 years for the first, long cycle period, but around 7 years on the average. The LL relations for the high-frequency and the low-frequency series are inverse to each other for 20-year periods (Figure 6a, bold lines). The dominant LL relation for the low-frequency series shows that NAO leads AMO pseudosignificantly 66% of the time. The multidecadal variability, and the leading role of NAO to the AMO is also studied by Nigam [16] (p. 5498). They use time series for AMO and NAO similar to ours, and they also LOESS(0.1)-smoothed the series as we did for the high-frequency component in Figure 5d [16]. These authors find that NAO leads the AMO by 16 years. A similar phase shift of ≈−15 years between the positive phases of NAO and AMO were also found by Sun and Li (p. 2, Figure 10, [3]) for their NAO+ and AMO/AMOC+ pair. Both estimates show a longer phase shift than we found. However, both studies estimate the pseudo-cycle period for AMO and NAO to be 50 to 70 years; thus, their phase shifts are $\frac{1}{4}$ of their cycle period, and in line with our estimate of 0.22 = 9.5/44 for the long cycle period.

The dominating clockwise rotation (NAO leading AMO) in the phase plot for the slightly LOESS(0.1)-smoothed low-frequency AMO(x) and the low-frequency NAO(y), support the role of NAO as a leading variable for the AMO.

### 5.5. The Ekman and the AMOC/NAO Relations

The high-frequency variance in the AMOC is dominated by Ekman transport anomalies and wind-stress-curl forcing. The winds over the North Atlantic Ocean vary at both seasonal and interannual and longer time scales, [14], thus affecting our high-frequency version of the AMOC time series. In line with Wang et al. (2019, p. 10) [14], we also find that the Ekman transport (Units Sv) correlated well with the NAO at close to zero time lag. Thus, the pressure difference expressed with the NAO translates into wind stress that again directly impacts sea-surface temperature.

### 5.6. The Bidecadal Periods

We stacked the absolute values for LL relations for the three high- and the three low-frequency paired series. There are four troughs in the stacked series around 1960, 1970, 1978, and 2000. Using time windows from 1960 to 1980, from 1980 to 2000, and from 2000 to 2020, we made a schematic summary of the LL relations between AMOC, AMO, and NAO (Figure 7). The relations suggest a causal relation, but do not prove causation. The arrows suggest leading relations; thicker arrows suggest stronger relations. The red arrows show leading directions that are persistent across time windows.

We find that for the high-frequency time series, AMO leads AMOC persistently across all three time windows. We obtain a circular sequence of AMO→AMOC→NAO→AMO during the period from 1980 to 2000. For the low-frequency series, NAO persistently leads AMO and AMO persistently leads AMOC across the three time windows: NAO→AMO→AMOC.

*5.7. Robustness*

We used data for the AMOC, AMO, and NAO from 1947 to present. The AMOC has been measured instrumentally from 2004. However, for all three series there are alternative series. The most common alternatives are to use different proxies for the series, or to emphasize different sets of months. Sun et al. (2021, p. 769) compared five different AMOC series estimated from different proxies and the raw AMO series and found that all AMOC series except one series led the AMO by 5 to 10 years during the period from 1955 to 2015.

Several series are represented only with the winter months, e.g., (D, J, F), or they are calculated across years, e.g., Atlantic Meridional Overturning Circulation (acsis.ac.uk), accessed 15 March 2022, report data from April to March. Although the two series most often show similar anomalies, the recent AMOC trend from 2009 to 2019 is negative for the annual Caesar data (R = −0.62), but positive for the cross-year Acis, UK data (R = 0.80), Appendix A. We have not managed to obtain information that would explain why the two AMOC representations show opposite slopes. The ocean variability series used here are retrieved from sources that are used abundantly in climate literature and we used annual data J to D., e.g., https://psl.noaa.gov/gcos_wgsp/Timeseries/ (accessed on 15 March 2022).

There are probably embedded series in the AMOC, AMO, and NAO series than we have identified. However, to identify such series with confidence, the series should be longer.

A second approach to evaluate robustness is to compare characteristics of the series that are independent, or where there are no obvious reasons why they should be dependent. Our results show that both cycle periods in the time series and cycle periods in their pairwise LL relations suggest cycle periods of about 20 years, supporting that both types of series express an inherent property of the ocean system. Third, Li et al. (2013, p. 5499) suggest relations between AMOC, AMO, and NAO that would be similar to our LL relations for the low-frequency patterns between 1960 and 1980. However, applying the high- resolution LL method to the data, we find that relations between the three climate variables change over time.

The method we use is relatively novel. To show that it gives reliable results, we applied it to the LOESS(0.3)-smoothed AMOC series, fixed at the observed dates and shifted 6 years forward. The results show that the fixed series is identified as a leading series, the cycle periods mirror the peak-to-peak distances in the series and the phase shift is 6 years on the average corresponding to the design phase shift, Appendix B. We believe we have established LL relations with high confidence that should give a good basis for causal interpretations.

*5.8. Future Work*

We have identified cycle periods over decadal, multidecadal, and up to half-century-long periods. Unfortunately, the time series are not long enough to identify longer cycle periods, and the additional anthropogenic warming of the globe may obliterate attempts to extend the time series into the future based on historic records.

However, disentangling the AMC, AMO, and NAO series into seasonal, interannual, multidecadal, and centennial series may allow closer studies of how seasonal mechanisms impact processes at increasingly longer time scales.

To further investigate the causal links between ocean variabilities, climate modeling techniques should be used. Unfortunately, as of 2020, the current models do not seem able to reproduce expressions of decadal or longer variabilities in the North Atlantic basin very well (Nigam and Sengupta [16] (p. 5502) and Smith and Scaife [5] (p. 799)).

**6. Conclusions**

The physical processes that give rise to the AMOC, the AMO, and the NAO time series must interact. We here show Lead–lag patterns between the series that would help to identify the interaction processes. We found that AMO→AMOC relation is persistent on an interannual time scale and the NAO→AMO→AMOC relation is persistent on a decadal

time scale. Apart from these two relations, the LL relations change sign at about 20-year periods. We have listed four mechanisms: compartment saturation, dynamic chaos and bifurcation, interaction between stochastic variability series and external "control knobs", that could contribute to the determination of cycle periods both for single-ocean series and for the LL relation between pairs of series, but it has not been possible to draw any conclusion. To examine the contribution from the four groups of mechanisms, one could apply the present high-resolution Lead–lag method to pairs of candidate causal and effect pairs to see if the cause comes before the effect.

**Author Contributions:** Conceptualization, K.L.S. and H.W.; methodology, K.L.S.; software, K.L.S.; validation, K.L.S. and H.W.; formal analysis, K.L.S.; investigation, K.L.S. and H.W.; resources, K.L.S.; data curation, H.W.; writing—original draft preparation, K.L.S.; writing—review and editing, K.L.S. and H.W.; visualization, K.L.S.; supervision, K.L.S.; project administration, K.L.S.; funding acquisition, K.L.S. All authors have read and agreed to the published version of the manuscript.

**Funding:** This research was funded by Oslo Metropolitan University to K.L.S., no grant number.

**Institutional Review Board Statement:** Not appliccable.

**Informed Consent Statement:** Not applicable.

**Data Availability Statement:** All data and all calculations are available from K.L.S.

**Acknowledgments:** We would like to thank Zeliang Wang for continuous support, helpful comments and advice and data support for the study. Furthermore, we would like to thank reviewers both for constructive criticisms of the present study and for suggesting interesting future work.

**Conflicts of Interest:** The authors declare no conflict of interest.

## Appendix A. AMOC Series

We have two time series for the AMOC over the period from 2004 to 2019. One series is from Caesar, McCarthy [6] and personal communication. The other is from the internet page http://www.acsis.ac.uk/climate-indicators/atlantic-meridional-overturning-circulation (accessed on 15 March 2022). The series are centered and normalized to unit standard deviation in Figure A1.

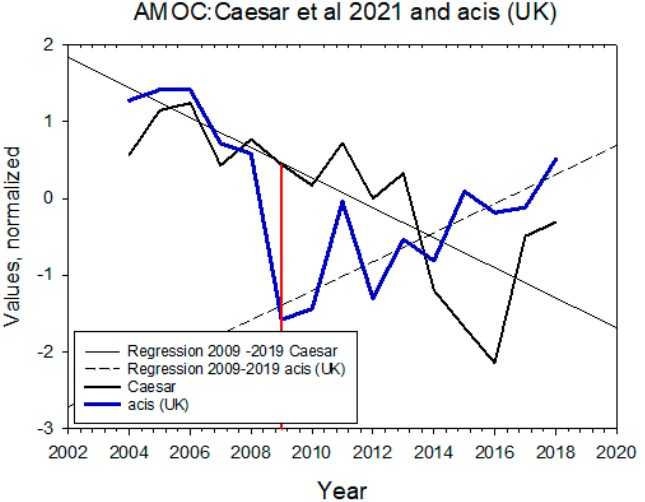

**Figure A1.** AMOC time series. The regression lines (straight lines) show regressions from 2009 to 2019; that is, the last decade. The Caesar, McCarthy [6] time series (black line) are annual averages. The blue line shows cross-year average values from March one year to April next year, and with the Ekman (wind-driven) component removed. The Acis (UK) values are read from a graph. The Caesar data were kindly supplied by Levke Caesar, Maynooth University.

Comparing the AMOC series with annual values and with only the winter months.

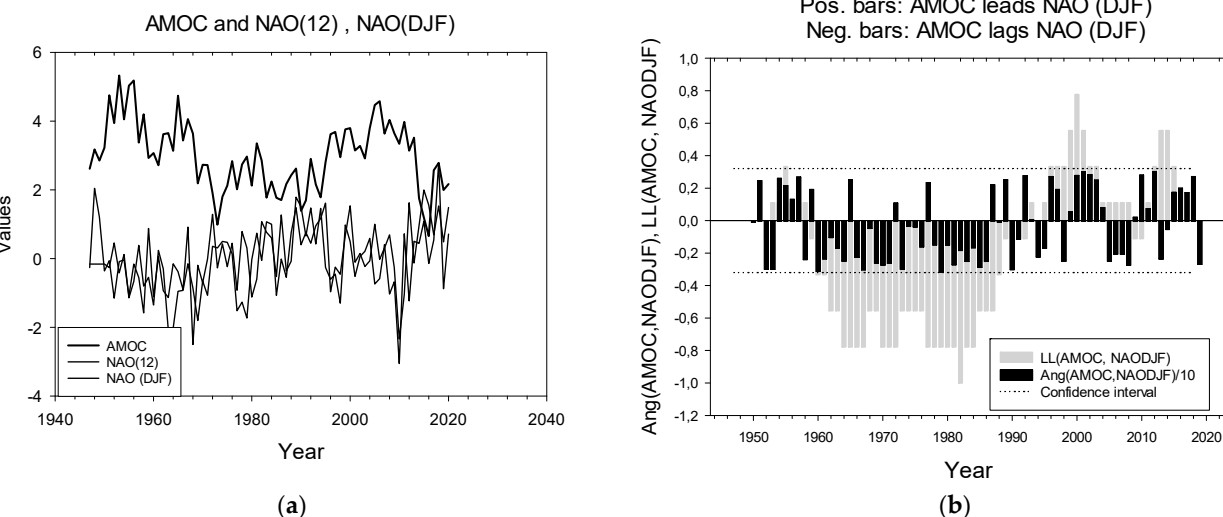

**Figure A2.** LL relations between AMOC and NAO (DJF). (**a**) Time series for AMOC and NAO (all months) and only the winter months (D, J, F); (**b**) LL relations. Note that the black bars are divided with 10 and are thus almost all significant. The period where AMOC lags NAO is from 1961 to 1988.

## Appendix B. Example

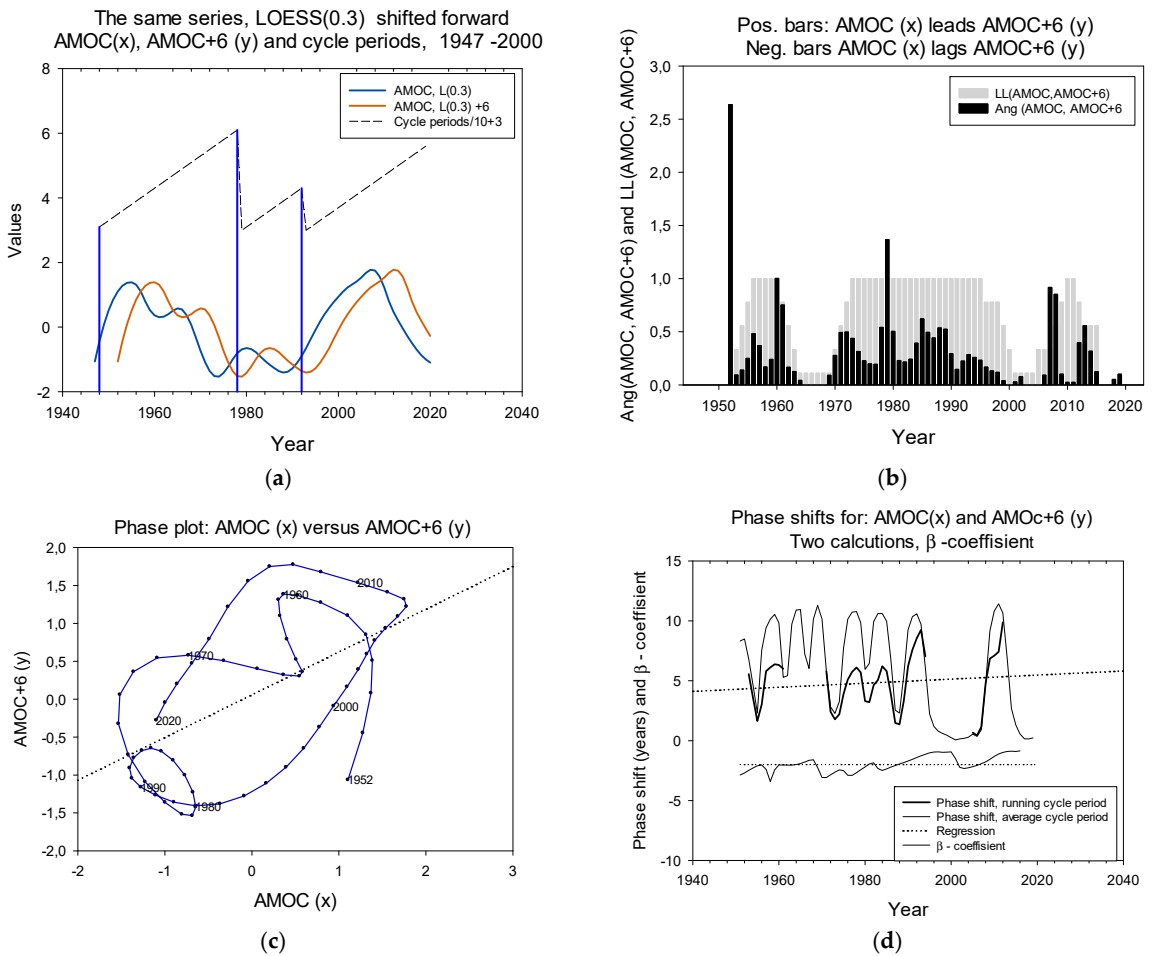

**Figure A3.** Example: AMOC LOESS(0.3)-smoothed (x) and shifted forward six years (y). (**a**) AMOC time series, original (blue) and shifted 6 years forward. The original annual AMOC series has been

LOESS(0.3)-smoothed. The zigzag curve indicates the length of years found by the cumulative angle method. (**b**) LL relations between AMOC original and AMOC shifted 6 years forward, LL(AMOC, AMOC+6). The black bars show Ang(3); that is, LL relation over three consecutive observations in the paired time series. The grey bars show LL(9); that is, the relation between positive and negative angles over 9 consecutive observations. (**c**) Phase plot for the pairs AMOC and AMOC+6. Note that most rotations are counterclockwise (positive, +) showing that AMOC leads AMOC +6. (**d**) Phase shifts calculated relative to moving cycle period and with average cycle period. Average cycle period is about 6 years, corresponding to the six years design phase shift. Lowe curve shows moving β-coefficient.

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
