# Peer review of "The North Atlantic Oscillations: Lead–Lag Relations for the NAO, the AMO, and the AMOC—A High-Resolution Lead–lag Analysis"

_climate, doi:10.3390/cli10050063_

Round 1

Reviewer 1 Report

In the work of the authors Seip, K.L., O. Green, and H. Wang, The North Atlantic Oscillation: Cycle Times for the NAO, the AMO and the AMOC. Climate, 2019. 729 7(3) the cycles of NAO, the AMO and the AMOS are considered. It was assumed that in this work the mechanisms causing the cycles of NAO, the AMO and the AMOC would be considered. However, the text and conclusions do not contain this information. There is information only about lead-lag patterns between the series.

The introduction describes the LL-method, which is more logical to move to the methods section. The introduction section should be supplemented with research by other authors.

Figure 3, 5 overlapping not all text is readable.

Figure 7 it is difficult to judge the predominance by the thickness of the arrow, it is better to represent it as a percentage.

Author Response

We use the acronym Rev. for the reviewers’ comments and Response for our response.

By some reasons, probably when we put the manuscript into the Climate format, part of the method section was lost. The lost section is not absolutely required,  but the method is easier to understand with that section present. The most important changes I the text are highlighted in bold. The arrows in Figure 7 change size when we close and open the manuscript. We will do the final editing in the final version of the manuscript.

Thank you for refereeing our work.  In particular, we appreciated your suggestions for improving Figure 7.

Rev. In the work of the authors Seip, K.L., O. Green, and H. Wang, The North Atlantic Oscillation: Cycle Times for the NAO, the AMO and the AMOC. Climate, 2019. 729 7(3) the cycles of NAO, the AMO and the AMOS are considered.

Response. This is a reference to a previous article where KLS was a coauthor. I see that one easily could infer from the title that it also would describe mechanisms that cause ocean variability, or ocean cycles, but that was not the intention. There is currently no consensus on what causes cycle periods, but several mechanisms have been proposed. We now list a set of possible mechanisms that could generate persistent (pseudo) cycles.

Rev. It was assumed that in this work the mechanisms causing the cycles of NAO, the AMO and the AMOC would be considered. However, the text and conclusions do not contain this information. There is information only about lead-lag patterns between the series.

Response. Sorry for the misleading. The intention was to examine persistence and regularity in lead- lag relations. To explore possible mechanisms that would cause  both persistence and regularities, we would suggest some modeling studies. However, to validate the models while they are developed, we believe that information on what LL- relations are realized is an important information.   

Rev. The introduction describes the LL-method, which is more logical to move to the methods section. The introduction section should be supplemented with research by other authors.

Response. A major idea for the article was to apply the high-resolution LL- method to the AMOC, AMO and NAO time series, therefore we thought it would be good to shortly present the method in the introduction section.

Rev. Figure 3, 5 overlapping not all text is readable.

Response. Sorry, Word manuscripts tend to be distorted when sending them by e-mail or other transfer systems. We have tried to make more space for the figures

Rev. Figure 7 it is difficult to judge the predominance by the thickness of the arrow, it is better to represent it as a percentage.

Response. Agree, we added the LL- strength to the figures, however, we kept the arrows because they give a quick overview of the LL- relations. (We also defined the LL-strength in the Figure text.)

Reviewer 2 Report

Review of the paper ‘The North Atlantic oscillations: Lead-lag relations for the NAO, the AMO and the AMOC. A high – resolution lead-lag analysis’ by Seip, K.L. and A., Wang, H., submitted to the Climate Journal.

The Authors present an extensive signal processing analysis of three key time series describing behavior of the Atlantic Ocean over decades. They use a relatively novel lead-lag technique, which is applied to the annual values of:

  • the North Atlantic Oscillation (NAO), reflecting pressure difference between a southern (Lisbon) and a northern station (Reykjavik),
  • the Atlantic Multi-decadal Oscillation (AMO), reflecting fluctuations of sea surface temperatures, and
  • the Atlantic Meridional Overturning Oscillation (AMOC), reflecting the current patterns of the Atlantic Ocean.

Although not stated explicitly, it can be deduced that the main objective of the manuscript was to identify lead-lag patterns among those three series. This is simultaneously the strongest and the weakest point of the paper. The strong aspect consists in the ability to identify clear lead-lag patterns in short time series and disentanglement of high and low frequency components. The weak one is that the identification of those patterns seems to be the aim it itself, without in-depth analysis of their background processes. Therefore, although the paper has a potential for publication in the Climate Journal, it should be amended to exploit this potential

The main lines of criticism are presented below:

  • Preoccupation on signal processing, clearly visible in the discussion and conclusion sections. The former contains detailed description of the characteristics of identified patterns without any references/hypotheses to possible background processes. This section could be acceptable if the background processes were covered in the conclusions. However, this section is very succinct - only 4 sentences, mostly highlighting the persistence of patterns that were identified. The Authors are strongly recommended to come up with a list of potential processes behind the patterns they identified.
  • One may suspect that the difficulties related to interpretation of the patterns from time series representing annual values prevented the Authors from any attempts to seek their physical explanations (background processes). Therefore, concentration on annual values seems problematic, because the annual values average out and thus obscure many physical processes, particularly the seasonal ones. On the other hand, the degree of obscuring shorter cycles could only be identified a posteriori. This could serve as one of the conclusions, directing future research to derivative series, e.g. representing seasonal behavior; the most obvious ones could be the winter index of NAO, AMO and AMOC derived from their Dec., Jan., Feb. and Mar. records. Such effects are very well imprinted in e.g. winter seawater variations of such a distant basin as the Baltic Sea, cf. Ocean Engineering, 109 (2015) 113–126. Consequently, trends in annual seasonality could be traced upon the examination of monthly indices, where the annual cycle is best pronounced. The Authors are recommend to explore the role of seasonal behavior and how it can be obscured by annual values.
  • Interpretation of mutual interactions, shown in Fig. 7 requires better structuring and explanation. The energy transfer predominantly runs from low to high density medium (air to water). Therefore, the NAO should be the primary driver of wave climates and one of the drivers of currents, water levels and other hydrodynamic quantities. Thus, it should control the AMOC series to a certain degree. Furthermore, both weather patterns, with the NAO as a proxy indicator, and the currents (with the AMOC as proxy indicator) should at least partly control sea surface temperature anomalies (AMO). However, such pattern emerged only for very long cycles (Fig. 7 lower panel). For shorter cycles we observe that AMO/AMOC lead the NAO, which appears unphysical, unless a credible explanation is provided. Here again we can see the preoccupation of the Authors on signal processing and difficulties in the interpretations of the outputs. They are recommended again to present a list of potential processes resulting in such lead-lag behavior in the context of energy transfer and its potential delayed effects.

Although the quality of English is good, the paper needs a thorough inspection. Several examples are listed below: 

L. 27: replace ‘is’ by ’are’ after ‘Neither’.

L. 115 replace ‘cure’ with ‘curve”.

L. 156 replace ‘ordinarl’ by ‘ordinary’.

L. 182 replace ‘coeffisient’ by ‘coefficient’ in the title of Fig. 3c

L. 287 titles of Fig. 4e,f not visible.

L. 326 titles of Fig. 5c,d not visible.

L. 468 double full stop.

L. 499 replace ‘are’ with ‘is’ at the end of that line.

L. 503 add ‘does’ after ‘that’.

L. 527 replace ‘are’ with ‘is’ after AMOC

L. 573 replace ‘shift’ with ‘shifts’.

L. 574 replace ‘year’ with ‘years’.

L. 601 replace ‘valuers’ with ‘values’.

L. 614 replace ‘form’ with ‘from’.

Author Response

We use the acronym Rev. for the reviewers’ comments and Response for our response.

By some reasons, probably when we put the manuscript into the Climate format, part of the method section was lost. The lost section is not absolutely required,  but the method is easier to understand with that section present. The most important changes I the text are highlighted in bold. The arrows in Figure 7 change size when we close and open the manuscript. We will do the final editing in the final version of the manuscript.

Thank you for reviewing our work. We very much appreciated the idea of making a list of possible background processes. We do not think it is possible to conclude (yet) on which process contributes to the lead-lag relations we identify but making the list and categorizing the possible processes are very useful.  The section that starts on line 573 outlines a list.

Rev.…Although not stated explicitly, it can be deduced that the main objective of the manuscript was to identify lead-lag patterns among those three series. This is simultaneously the strongest and the weakest point of the paper. The strong aspect consists in the ability to identify clear lead-lag patterns in short time series and disentanglement of high and low frequency components. The weak one is that the identification of those patterns seems to be the aim it itself, without in-depth analysis of their background processes. Therefore, although the paper has a potential for publication in the Climate Journal, it should be amended to exploit this potential

Response. We should have stated explicitly that the main objective of the study was to establish high-resolution lead-lag relations between the high and low frequency components of the North Atlantic oscillations (or variability series; but we apply power spectral density algorithms to the series implying that the series are superpositions of cyclic series).  We now add a short paragraph in the introduction explaining that there are not many (if any) studies that conclude with confidence about the background processes for LL- relations between ocean oscillation series. However, following the referee’s advice, we list and shortly discuss hypotheses in the literature for why periodicity occur in climate time series. We were tempted to suggest some concluding remarks on which background processes are most probable but decided it would be too speculative at the present time.

Rev. The main lines of criticism are presented below:

  • Preoccupation on signal processing, clearly visible in the discussion and conclusion sections. The former contains detailed description of the characteristics of identified patterns without any references/hypotheses to possible background processes. This section could be acceptable if the background processes were covered in the conclusions. However, this section is very succinct - only 4 sentences, mostly highlighting the persistence of patterns that were identified. The Authors are strongly recommended to come up with a list of potential processes behind the patterns they identified.

Response.  Thank you, we very much liked your suggestion. Although there is no consensus on the mechanisms that would cause periodicities, e.g., Sun et al. (2021, p. 240) on the AMO, there are several suggested mechanisms. We now add a list of possible mechanisms and suggest methods for assess the likelihood that the mechanisms could generate the LL- patterns we observe.

Rev. One may suspect that the difficulties related to interpretation of the patterns from time series representing annual values prevented the Authors from any attempts to seek their physical explanations (background processes). Therefore, concentration on annual values seems problematic, because the annual values average out and thus obscure many physical processes, particularly the seasonal ones.

Response. The use of seasonal values for ocean time series is an interesting study objective. In many cases, winter values are used, for example as Dec, Jan. Feb. March, or some other restrictions on months to be included. It also appears that the winter values are the most important values to use to detect physical processes that act on the system. However, in the present study we focused on interactions between AMOC, AMO and NAO on interannual and decadal time scales. We now compare results on the LL-relations between AMOC and NAO based on annual values and on winter values (D,J,F).  The results for the winter values appear to be clearer (more significant LL- relations) than the results for annual values, but the results are similar, and we do not have to make the selection of months to include.

But we agree with the reviewer that  directing future research to the effects from seasonal behavior is a very good idea. Doing this may clarify the interaction between seasonal, interannual and decennial processes, as well as the seasonality of the interactions at the interannual and decennial timescales..   

Rev. On the other hand, the degree of obscuring shorter cycles could only be identified a posteriori. This could serve as one of the conclusions, directing future research to derivative series, e.g., representing seasonal behavior; the most obvious ones could be the winter index of NAO, AMO and AMOC derived from their Dec., Jan., Feb., and Mar. records. Such effects are very well imprinted in e.g., winter seawater variations of such a distant basin as the Baltic Sea, cf. Ocean Engineering, 109 (2015) 113–126. Consequently, trends in annual seasonality could be traced upon the examination of monthly indices, where the annual cycle is best pronounced. The Authors are recommended to explore the role of seasonal behavior and how it can be obscured by annual values.

Response. Thank you, we agree with your suggestions for future research, and we have consulted the article that you recommend. Although annual values may obscure the information in seasonal variability, we think an additional interesting issue would be to investigate if changes in seasonal development would translate into changes on annual scales, and further, whether changes on annual scales translate into changes on decadal scales. 

Rev. Interpretation of mutual interactions, shown in Fig. 7 requires better structuring and explanation.

Response. Thank you. We now add the numerical values for LL- strength in Figure 7 and explained better how LL- strength is defined , Eq.(2). Furthermore, in the previous figure we used the LL- strength of the three midpoint dates. We now use the average LL- strength values for the 10 midpoint years.

(The arrows in the figure change thickness when we close and open the manuscript. We will do the final adjustments in the final version of the manuscript)

Rev. The energy transfer predominantly runs from low to high density medium (air to water). Therefore, the NAO should be the primary driver of wave climates and one of the drivers of currents, water levels and other hydrodynamic quantities. Thus, it should control the AMOC series to a certain degree.

Response. The energy transfer direction you sketch is the major directions we find in our study. Your comments and suggestions  are very helpful. However, you also add “to a certain degree”  and that seems to be the case, in particular for NAO/AMOC pair. We also find that the transfer differs in strength over time. We would assume that transfer strength also differs with season,  confirming your suggestion that studying time series that are restricted to different seasons would be interesting.  For example, would the series of summer months give patterns that resemble the annual patterns?

Ref. Furthermore, both weather patterns, with the NAO as a proxy indicator, and the currents (with the AMOC as proxy indicator) should at least partly control sea surface temperature anomalies (AMO).

Response. We find that NAO leads AMO that again leads AMOC at the low frequency time scale. However, NAO both leads and lags AMOC. At the high frequency time scale NAO both leads and lags AMO and NAO both leads and lags AMOC. AMO always leads AMOC, and the  strength increases with time.

Ref. However, such pattern emerged only for very long cycles (Fig. 7 lower panel). For shorter cycles we observe that AMO/AMOC lead the NAO, which appears unphysical, unless a credible explanation is provided. Here again we can see the preoccupation of the Authors on signal processing and difficulties in the interpretations of the outputs. They are recommended again to present a list of potential processes resulting in such lead-lag behavior in the context of energy transfer and its potential delayed effects.

Response. AMOC leads NAO during periods both in the  high frequency and the low frequency domains. We do not have any firm explanation for this, but we do not see that it must be unphysical. The NAO expresses sea level pressure differences, and those differences could be a response to temperature changes in the AMO (and AMOC and AMO covary tight so it is difficult to distinguish between the effects from the series.)

And again, thank you for suggesting us to make a list of potential processes that could result in lead- lag behavior. However, we try to be careful stretching the reasoning too far based on the data available. We believe an in-depth study can be best made through a modeling study but using the results in the present study to calibrate and evaluate the model.    

Rev. Although the quality of English is good, the paper needs a thorough inspection. Several examples are listed below:  Suggestions were added here.

Response. We accepted all the suggested corrections. Thank you. We will ask the journal to do a “language washing” of the manuscript.  

Sun, C., Y. Liu, J. Xue, F. Kucharsk, J. LI and X. Li (2021). "The importance of inter‐basin atmospheric teleconnection in the SST footprint of Atlantic multidecadal oscillation over western Pacifc." Climate dynamics 57: 239-252.

Round 2

Reviewer 2 Report

You have improved your work, however there is still a feeling you paid most attention to signal processing and much less to the interpretation of results. Anyway, the paper is ready for publishing and should serve as background for follow-up studies. 

A linguistic correction: L. 557 replace 'fond' with 'found'.